# Machine learning-based urban noise appropriateness evaluation method and driving factor analysis

Jinlin Teng[1,2], Cheng Zhang[1,2], Huimin Gong[1,2], Chunqing Liu[1,2]*

1 College of Forestry, Jiangxi Agricultural University, Nanchang, Jiangxi Province, China, 2 Jiangxi Rural Cultural Development Research Center of Jiangxi Agricultural University, Nanchang, Jiangxi Province, China

* 1397402332@qq.com

**Data Availability Statement:** All Noise data files are available from the GitHub database(Get website address:https://github.com/tengjinlin/noise-prediction.git); Other relevant data can be found in the manuscript and its supporting information files.

## Abstract

The evaluation of urban noise suitability is crucial for urban environmental management. Efficient and cost-effective methods for obtaining noise distribution data are of great interest. This study introduces various machine learning methods and applies the Random Forest algorithm, which performed best, to investigate noise suitability in the central urban area of Nanchang City. The findings are as follows:

1.Machine learning algorithms can be effectively used for urban noise evaluation. The optimized model accurately reflects the noise suitability levels in Nanchang City.

2.The feature importance ranking reveals that population spatial distribution has the most significant impact on urban noise, followed by distance to water bodies and road network density. These three features significantly influence the assessment of urban noise suitability and should be prioritized in noise control measures.

3.The weakly suitable noise areas in Nanchang's central urban region are primarily concentrated on the east bank of the Ganjiang River, making this a key area for noise management. Overall, the Unsuitable, Slightly suitable, Moderately suitable, Relatively suitable, and Height suitable areas constitute 9.38%, 16.03%, 28.02%, 33.31%, and 13.25% of the central urban area, respectively.

4.The SHAP model identifies the top three features in terms of importance, showing that different values of feature variables have varying impacts on noise suitability.

This study employs data mining concepts and machine learning techniques to provide an accurate and objective assessment of urban noise levels. The results offer scientific decision-making support for urban spatial planning and noise mitigation measures, ensuring the healthy and sustainable development of the urban environment.

## 1 Introduction

With significant achievements in the battles against air, water, and soil pollution, residents' tolerance for urban environmental issues has continued to decrease. Noise pollution has gradually become a major focus of environmental complaints and is one of the current weaknesses

**Funding:** The author(s) received no specific funding for this work.

**Competing interests:** I have read the journal's policy and the authors of this manuscript have the following competing interests: The authors have declared that no competing interests exist.

in ecological and environmental protection [1–3]. This is reflected in the "2020 National Public Service Quality Monitoring Report" released by the State Administration for Market Regulation in 2021, where "satisfaction with surrounding noise control" ranked last among all 68 evaluation indicators. Additionally, the "2023 China Noise Pollution Prevention Report" released by the Ministry of Ecology and Environment of China indicated that noise pollution accounted for 59.9% of all environmental pollution complaints [4]. Therefore, strengthening urban noise management is imperative.

Noise, as an environmental pollutant, has increasingly been recognized by the academic community for its harmful effects on human health. Although noise research has a long history, it has not received the same level of attention as other pollutants, as evidenced by the quantity and quality of related literature. Noise pollution remains a major environmental health issue in many countries, including China. Previous studies have shown that, in addition to affecting the auditory system [5], noise can cause annoyance, disrupt sleep, and impair cognitive abilities [6]. Physiologically, noise pollution can lead to systemic changes—prolonged exposure to high noise levels can damage hearing, and harm the cardiovascular [7, 8], nervous [9], and endocrine systems [10, 11]. Therefore, understanding the extent of noise pollution, its impact on human health, and ways to mitigate these effects is urgent [12, 13]. Research on urban noise suitability is crucial. Scholars have explored various aspects of urban acoustic environments:

1. Optimizing the acoustic environment of major urban functional areas, such as residential [14], commercial [15], transit [16, 17], and industrial zones [18], to improve urban noise conditions.

2. Highlighting the role of various types of urban green spaces in noise control. Urban parks [19, 20], open green spaces [21, 22], and traffic greenbelts [23] are effective noise reduction measures.

3. Emphasizing the positive impact of soundscape elements on noise pollution. Natural sounds like water [24] and bird songs [25] can effectively mitigate urban noise by masking artificial sounds and enhancing residents' psychological comfort.

Currently, an effective and intuitive method in the academic field for assessing and controlling noise pollution in urban environments is the use of noise maps. These maps visually display noise distribution, aiding in the formulation of precise mitigation measures to effectively reduce noise pollution. In 2002, the European Union issued the Environmental Noise Directive (2002/49/EC), requiring member states to create noise maps to assess and manage noise exposure from major roads, railways, airports, and agglomerations [26].

Most existing noise maps are based on traditional sound propagation algorithms, which simulate the paths of sound waves through the air to calculate sound levels at various points, eventually visualizing the noise distribution [27]. Some researchers collect noise data using static recording stations distributed throughout the city or through mobile measurements that gather a large number of geographically tagged audio recordings [28, 29]. The noise data between multiple sample points can be mapped using methods such as co-kriging interpolation. However, interpolated data sometimes fail to accurately reflect actual noise levels in certain local areas. Interpolation methods consider the spatial correlation of data and can sometimes yield accurate predictions. However, if sample data are unevenly distributed or contain outliers in specific areas, the interpolation results can be significantly affected. This paper proposes a machine learning-based solution for urban noise assessment.

This approach relies on training with large datasets, allowing it to automatically extract features and patterns from the data. As a result, even if interpolation fails in certain areas, the

machine learning model can still predict actual noise levels by learning from data in other regions. This approach does not heavily rely on direct measurements of noise sources. Instead, it utilizes existing urban noise and geospatial data to identify potential patterns and trends and applies machine learning algorithms to predict noise levels across the entire urban area. This method minimizes the impact of urban artificial disturbances and can serve as a cost-effective, efficient, and accurate approach to objectively reflect urban noise assessment levels. It provides detailed characteristics of urban noise spatial distribution and identifies the main factors driving these patterns.

## 2 Materials and methods

### 2.1 Study area

This study focuses on Nanchang, the capital of Jiangxi Province. The noise environment in Nanchang is representative of those in other provincial capitals in China. Like many medium to large cities, Nanchang faces significant population pressure, resulting in common issues such as traffic noise, industrial noise, construction noise, and social noise. Therefore, Nanchang's noise environment serves as a typical example for similar urban areas. The Environmental Protection Bureau's monitoring data in Nanchang is concentrated along the Ganjiang River, covering five districts: Donghu, Xihu, Qingyunpu, Qingshanhu, and Honggutan. Therefore, our research area encompasses these five districts, totaling 585 square kilometers with a population of 2.6 million.

### 2.2 Data sources

**2.2.1 Data acquisition.** This study utilizes noise monitoring data to construct sample categories. The data comes from the annual environmental noise quality monitoring conducted by China, specifically sourced from the Nanchang Environmental Protection Bureau's Environmental Quality Monitoring Database (http://sthjj.nc.gov.cn/ncgbj/zsjcbg/nav_list.shtml). The research area includes 103 monitoring points, each with a clearly defined location name. The spatial location of each noise monitoring point was determined using coordinate collectors (Fig 1). The noise monitoring methods comply with national standards, including the "Technical Specifications for Environmental Noise Monitoring—Urban Acoustic Environment Routine Monitoring" (HJ 640–2012) and the "Environmental Quality Standards for Noise" (GB3096-2008). The measured data includes equivalent sound level data from 2021, 2022, and 2023. Additionally, the data includes traffic volumes of large vehicles, defined by the "Road Traffic Management—Motor Vehicle Types" (GA802-2019) as passenger cars with a length ≥6m or carrying ≥20 passengers, and freight vehicles and trailers with a total mass ≥12t.

Considering the urban noise limits specified in China's "Environmental Quality Standards for Noise" and the research by Kangjian [30], Ising [31], and others, a threshold of 65dB was used for data classification in the machine learning algorithm.

**2.2.2 Feature matrix construction.** Urban noise is influenced by multiple factors, and the construction of feature indicators lacks a systematic approach. Therefore, when selecting feature factors, it is essential to comprehensively consider their independence, avoiding highly correlated features while ensuring thoroughness. Additionally, to ensure the scientific validity and persuasiveness of the feature matrix, the selected factors should be well-validated and recognized in the field of urban noise research, encompassing multiple dimensions. Based on previous studies on urban noise, this paper summarizes the feature factors into two main categories:

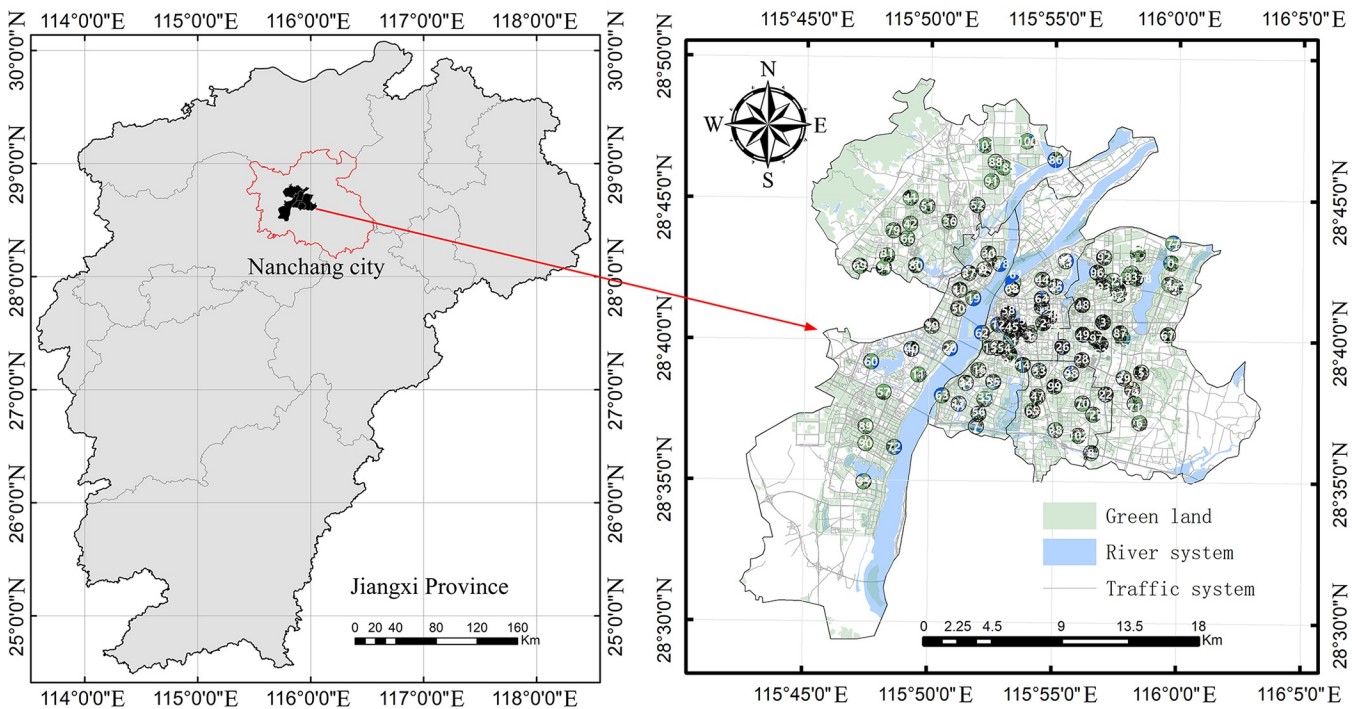

**Fig 1. Study area and noise monitoring points.**

1. Natural Environmental Conditions

The natural environment plays a significant role in regulating and influencing noise through absorption, attenuation, masking, and improvement effects. Key aspects include terrain [32], vegetation, wind speed [33], water systems, species diversity, and environmental purification. The following indicators were selected to reflect these influences:

◆ Elevation

◆ Normalized Difference Vegetation Index (NDVI)

◆ Annual average wind speed

◆ Distance to water bodies

◆ Biodiversity value

◆ Environmental purification value

2. Urban Development Conditions

Urban development conditions significantly impact the distribution of urban noise, particularly in high-density urban areas. Factors such as land use, population density, resident activities, traffic flow [34], road layout, and the relative position and height of buildings [35] directly influence urban noise distribution. The following indicators were selected to reflect these influences:

◆ Land use

◆ Population distribution

◆ Distance to residential areas

**Table 1. Selection of noise suitability characteristics in the central urban area of Nanchang.**

| Influence Factors | Feature Selection | Feature Description | Calculation Method |
|---|---|---|---|
| Natural Environmental Conditions | Terrain | Affects noise propagation paths and attenuation levels | DEM value |
| | NDVI | Reflects vegetation cover in the area, effectively blocking noise propagation and masking artificial noise | Derived from remote sensing image bands: NIR (near-infrared)—IR (infrared band) / NIR + IR |
| | Wind Speed | Affects noise diffusion and is a major urban noise source | Obtained from published datasets |
| | Distance to Water Bodies | Affects noise propagation and diffusion | ArcGIS distance analysis tool, using Euclidean distance |
| | Biodiversity | Higher biodiversity enhances noise suitability | Obtained from published datasets |
| | Environmental Purification Value | Affects noise propagation | Obtained from published datasets |
| Urban Development Conditions | Land Use | Reflects urban land spatial structure and land use characteristics | Obtained from existing datasets |
| | Population Distribution | Reflects the number of people in the area | Obtained from existing datasets |
| | Distance to Residential Areas | Reflects the intensity of residential activities; smaller values indicate more frequent activities | ArcGIS distance analysis tool, using Euclidean distance |
| | Large Vehicle Traffic Volume | Reflects traffic volume that directly impacts urban noise | Nanchang Environmental Protection Bureau's environmental quality monitoring database, interpolated in GIS |
| | Road Network Density | Reflects urban road layout; higher values indicate greater impact on noise | Calculated using ArcGIS |
| | Building Height | Affects noise propagation | Obtained from existing datasets |

♦ Large vehicle traffic volume

♦ Road network density

♦ Building height

From these two aspects, natural environment and urban development conditions, relevant feature factors were selected to construct the noise suitability feature matrix for the central urban area of Nanchang (Table 1). The sources of the data are listed in Table 2.

## 2.3 Machine learning modeling methods

Machine learning uses nonlinear fitting to establish relationships between feature variables and known sample categories. This modeling process enables the prediction of new sample classifications using the constructed model. This approach can automatically learn and identify complex patterns and features in the data. Through training and model optimization, it achieves accurate classification and prediction of unknown data, improving efficiency and accuracy while reducing labor costs. Additionally, it helps uncover hidden patterns and trends in the data, providing more scientific decision-making support. To determine the most suitable machine learning algorithm for the noise data in Nanchang's central urban area, we trained and evaluated six commonly used classification algorithms: Decision Trees (DT), Random Forest (RF), Support Vector Machine (SVM), Logistic Regression(LR), K-Nearest Neighbors (KNN), and Gradient Boosting(GB).

Decision Trees (DT) are tree-based algorithms used for classification and regression tasks [36]. They construct a tree structure by dividing the dataset into subsets, with each node representing a feature, each branch representing a feature value, and the final leaf nodes representing a class (or regression value). Decision trees are easy to understand and interpret, and they can handle both numerical and categorical data, but they are prone to overfitting [37].

**Table 2. Data sources.**

| Data Type | Data Year | Data Source | Data Description |
|---|---|---|---|
| Land Use Data | 2020 | 30m Global Land Cover Data (http://www.globallandcover.com/) | 30m spatial resolution, reflecting land cover conditions |
| Population Data | 2020 | WORLD POP Dataset (www.worldpop.org/) | Spatial resolution of 3 arc seconds, approximately 100m at the equator |
| Remote Sensing Data | 2020 | USGS Remote Sensing Data (https://earthexplorer.usgs.gov/) | 30m Landsat 8 OLI_TIRS data used to calculate the Normalized Difference Vegetation Index (NDVI) |
| Ecosystem Service Value Data | 2020 | Spatial Distribution Dataset of China's Terrestrial Ecosystem Service Value (http://www.resdc.cn/DOI) | Biodiversity and environmental purification value data |
| Meteorological Data | 2020 | Annual Spatial Interpolation Dataset of Chinese Meteorological Elements (http://www.resdc.cn/DOI) | Annual average wind speed (WIN) data |
| Basic Vector Data | 2020 | https://www.openstreetmap.org | Used to calculate road network density and distance to water bodies using GIS kriging method |
| Building Height Data | 2020 | Zenodo Platform Database (https://zenodo.org/records/7827315) | 10m resolution building height raster data |
| Residential POI Data | 2023 | Baidu Map API Open Interface | 5,243 residential points obtained after data cleaning; distance to residential areas calculated using GIS kriging method |
| Noise Monitoring Data | 2021 2022 2023 | Nanchang Environmental Protection Bureau Environmental Quality Monitoring Database (http://sthjj.nc.gov.cn/ncgbj/zsjcbg/nav_list.shtml) | Includes equivalent sound level and large vehicle traffic volume data for 103 monitoring points within the study area |

Random Forest (RF) is an ensemble learning algorithm that improves the accuracy of classification and regression tasks by training multiple decision trees. It enhances model performance by averaging or voting on the predictions of numerous decision trees, reducing overfitting [38]. RF is suitable for large datasets, robust to outliers and missing data, and can estimate feature importance.

Support Vector Machine (SVM) is a supervised learning algorithm used for binary classification and regression tasks. It finds the optimal hyperplane in the feature space to separate data points from different classes, maximizing the margin between them [38]. SVM has highly flexible kernel functions, can handle high-dimensional datasets, and is robust to outliers, but it requires longer training times for large datasets.

Logistic Regression is a linear model used for binary classification problems. It classifies based on a linear combination of features, using the logistic function to convert the linear output into probabilities, which are then used for classification based on a threshold. Logistic regression is simple and easy to implement, can output class probabilities, but has limited ability to model non-linear problems.

K-Nearest Neighbors (KNN) is an instance-based learning algorithm used for classification and regression tasks. It classifies by considering the class of the nearest neighbors to the input instance, using voting to determine the class of a sample. KNN requires no model training, is easy to understand and implement, but has slower prediction speed for large datasets and is less suitable for high-dimensional data.

Gradient Boosting is an ensemble learning algorithm that improves model performance by sequentially training multiple weak learners. It iteratively trains new models to correct the errors of the previous model, gradually enhancing model performance. Gradient boosting generally performs well in prediction tasks, but it is sensitive to hyperparameter selection and tuning, requiring some experience in hyperparameter optimization [37].

## 2.4 Experimental procedure

The data processing and modeling procedure in this study is as follows:

1. Data Aggregation and Preprocessing: Multiple sources of geospatial data were aggregated. The data were preprocessed according to the geographic factors represented by the feature indicators to construct the feature matrix (Fig 2).

2. Feature Extraction and Data Split: The feature values corresponding to the sample points were extracted. The sample data were divided into a training set (60%, 185 samples) and a test set (40%, 124 samples). The training set was used to train the model in Python, and the test set was used for validation.

3. Model Parameter Adjustment: Model parameters were adjusted and various classification evaluation metrics were validated using 10-fold cross-validation, confusion matrices, ROC curves, and learning curves. Model parameters were fine-tuned to achieve relatively optimal accuracy.

4. Visualization of Noise Suitability Evaluation: The entire urban area was divided into 58,508 grids of 100m×100m. The feature matrix of these grids was input into the trained model for classification prediction. The results were visualized to display the noise suitability map and conduct relevant analysis.

## 3 Results and analysis

### 3.1 Model reliability evaluation

The confusion matrix is a commonly used tool in machine learning, particularly for classification problems. It summarizes the performance of a model by highlighting its accuracy. From the confusion matrix, we can calculate metrics such as precision, recall, and the F1 score to evaluate the model's performance across different categories.

The Receiver Operating Characteristic (ROC) curve is a graphical tool used to assess the performance of binary classification models. It shows the relationship between the true positive rate (also known as sensitivity) and the false positive rate (1-specificity). The area under the ROC curve (AUC) provides a measure of the model's performance across different thresholds. A higher AUC value, closer to 1, indicates better model performance.

Learning curves are used to evaluate how a machine learning model's performance changes as the number of training samples increases. They also assess the model's performance during cross-validation. Learning curves help identify whether the model benefits from additional training data and indicate whether the model is overfitting or underfitting. Based on this, we use a combination of confusion matrices, ROC curves, and AUC scores to evaluate model classification performance, alongside learning curves to assess overfitting or underfitting.

In the formulas:TP (True Positive) represents the number of samples that are actually positive and correctly predicted as positive.FP (False Positive) represents the number of samples that are actually negative but incorrectly predicted as positive.FN (False Negative) represents the number of samples that are actually positive but incorrectly predicted as negative.The F1 score is the harmonic mean of precision and recall. A higher F1 score indicates a more robust model.

In this study, we used Python's "plot_learning_curve" function to draw learning curves. We performed 10-fold cross-validation to calculate the F1 scores at 10 equidistant points from 10% to 100% of the training data size. Ideally, as the number of training samples increases, the training score should gradually decrease because the model needs to generalize to more samples, while the cross-validation score should increase, indicating that the model learns more from additional data. If the training score is very high but the cross-validation score is low, it may indicate overfitting, where the model performs well on training data but poorly on unseen data. If both scores are low, the model might be underfitting.

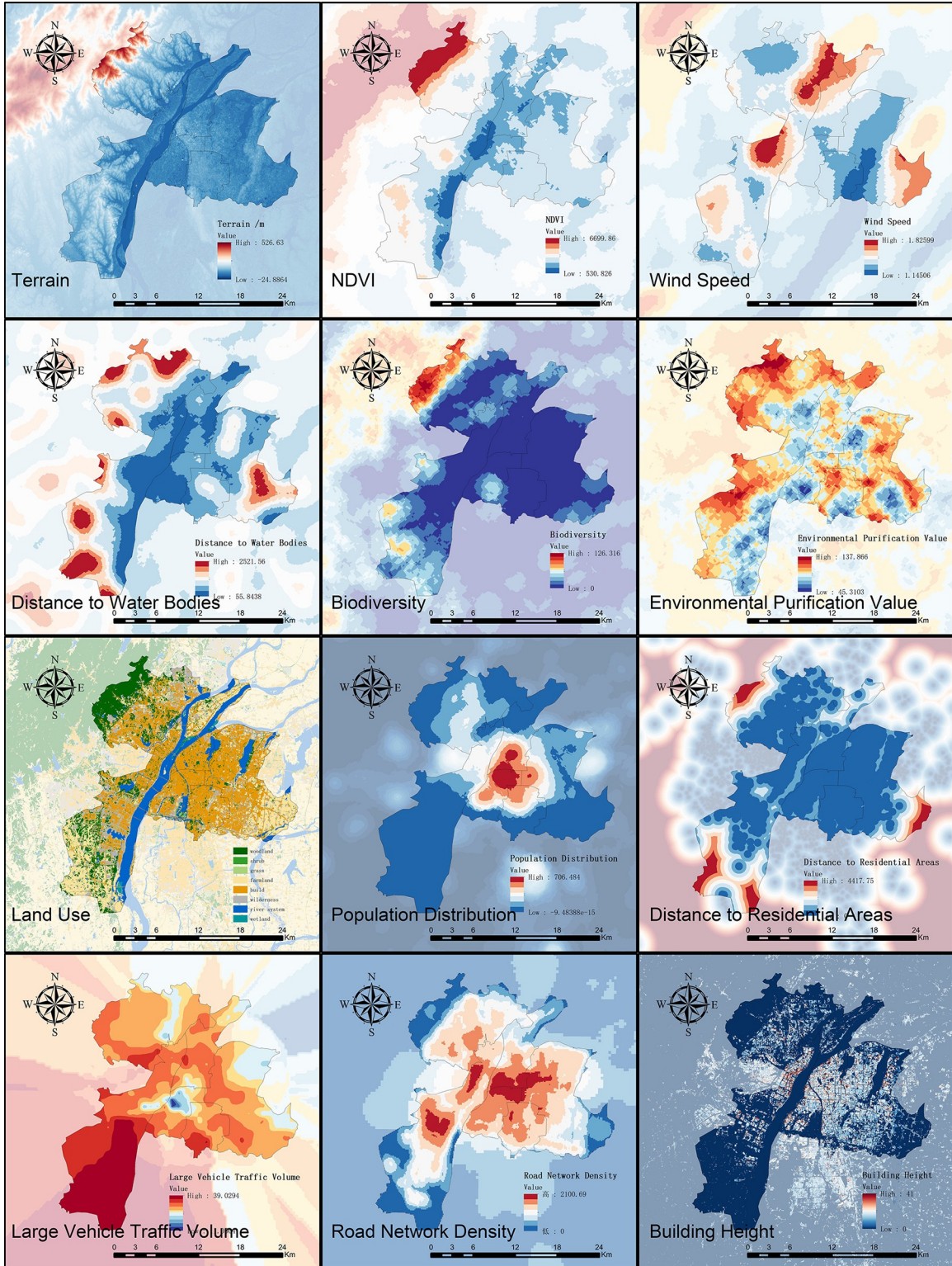

**Fig 2. Characteristics of urban noise suitability factors.**

**Table 3. Model classification evaluation indicators under various machine algorithms.**

| Model | Precision | | Recall | | F1 Score | | AUC |
|---|---|---|---|---|---|---|---|
| | Test | Training | Test | Training | Test | Training | |
| DT | 0.838710 | 0.981132 | 0.838710 | 0.859504 | 0.838710 | 0.916300 | 0.87 |
| RF | 0.814286 | 0.918033 | 0.919355 | 0.925620 | 0.863636 | 0.921811 | 0.89 |
| SVM | 0.636364 | 0.812950 | 0.790323 | 0.933884 | 0.705036 | 0.869231 | 0.79 |
| LR | 0.571429 | 0.724832 | 0.838710 | 0.892562 | 0.679739 | 0.800000 | 0.74 |
| KNN | 0.620253 | 0.824427 | 0.790323 | 0.892562 | 0.695035 | 0.857143 | 0.72 |
| GB | 0.771429 | 0.898438 | 0.870968 | 0.950413 | 0.818182 | 0.923695 | 0.88 |

The results show that different machine learning algorithms exhibit varying levels of accuracy with our data, as detailed in Table 3 and visualized in Fig 3. Among the six machine learning algorithms, Decision Tree, Random Forest, and Gradient Boosting algorithms performed exceptionally well, with relatively high AUC values as shown in Fig 4. Conversely, Support Vector Machine, Logistic Regression, and K-Nearest Neighbors algorithms performed poorly. The learning curves in Fig 5 indicate that the best models have high training and cross-validation scores with minimal differences, suggesting good performance and generalization. Random Forest and Gradient Boosting algorithms, in particular, showed smooth curves and high scores, indicating better stability and generalization.

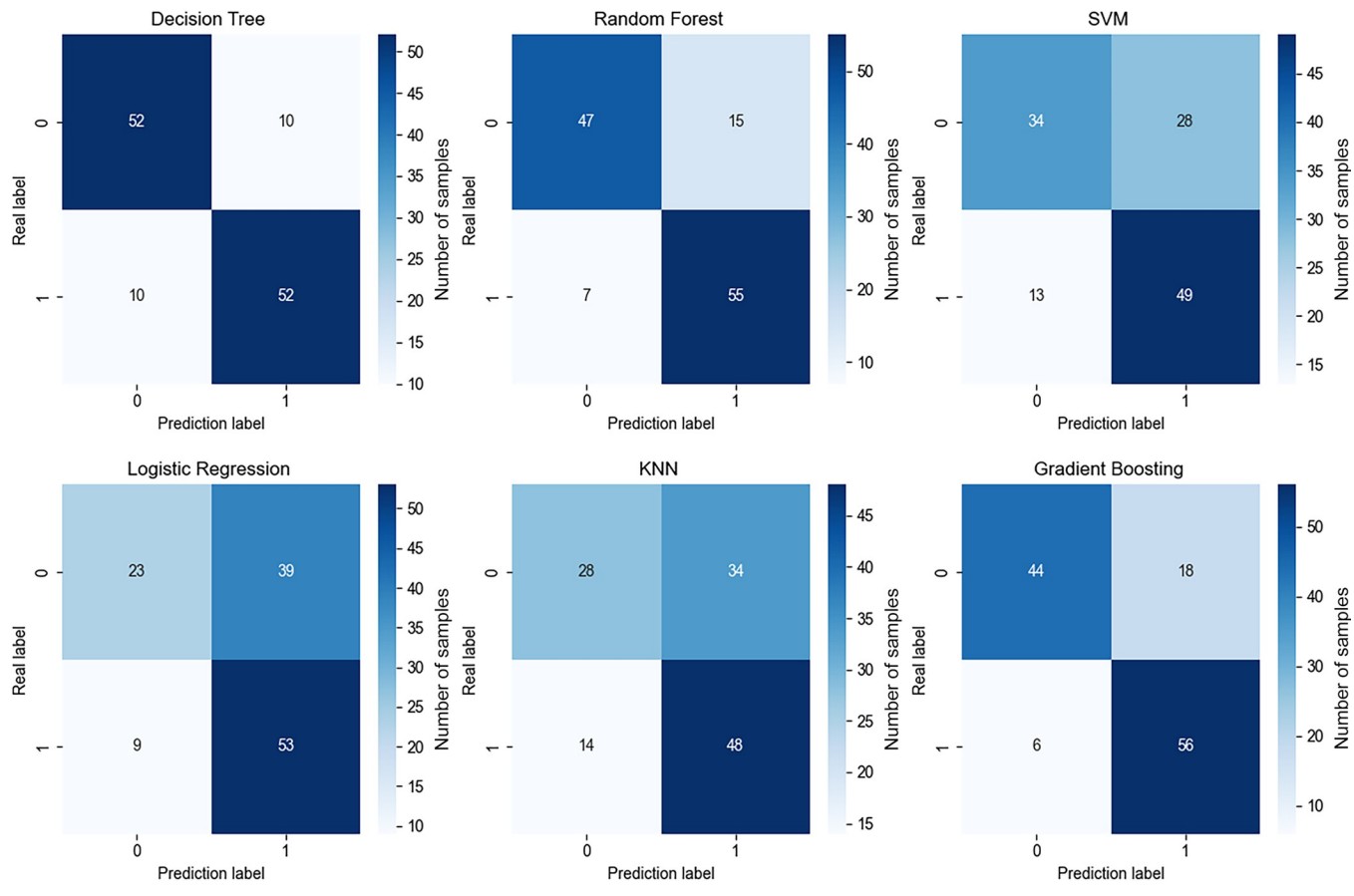

**Fig 3. Confusion matrices under various machine algorithms.**

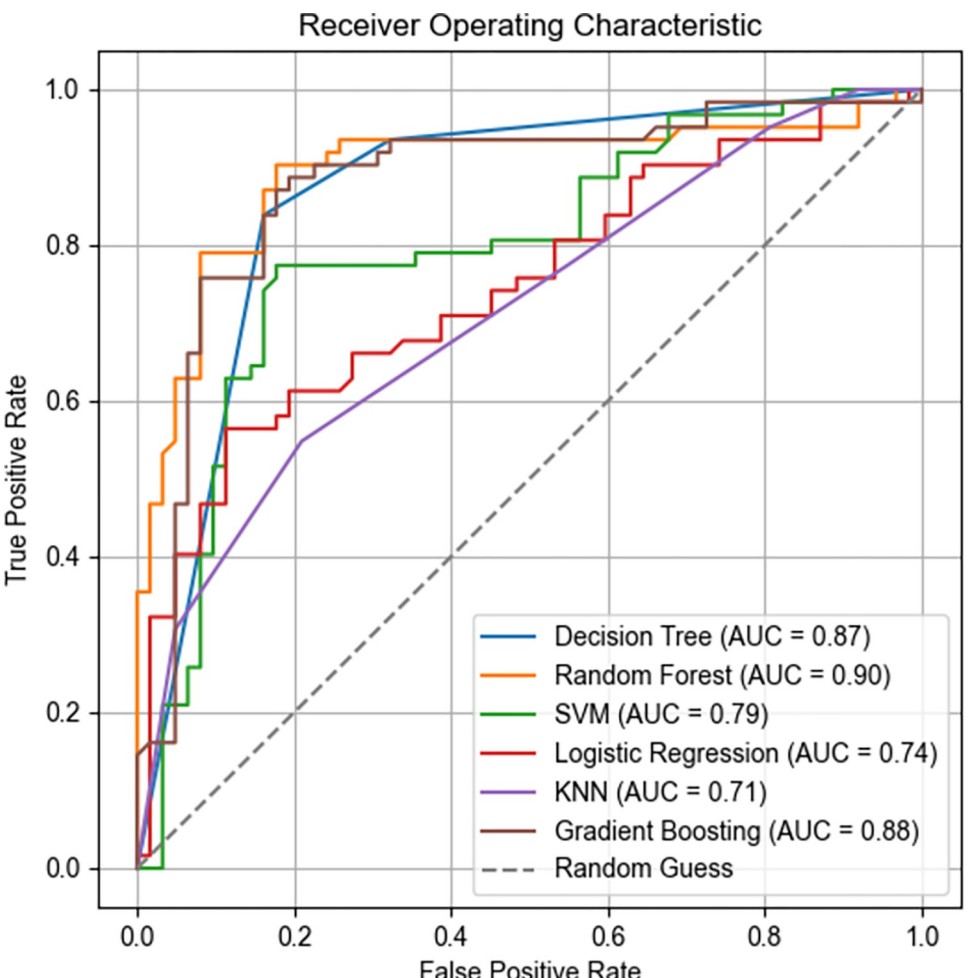

**Fig 4. ROC curves under various machine algorithms.**

## 3.2 Urban noise suitability analysis

Based on the urban noise scenarios and the research requirements, we selected the Random Forest model, which performed the best (Test F1 Score = 0.8636; Training F1 Score = 0.9218; AUC = 0.90), for prediction. We divided the entire urban area into 58,508 grids of 100m×100m and input the feature matrix data of these grids into the trained model to obtain the noise suitability prediction probabilities. Using the natural breaks method for visualization, the results were categorized into five classes: Height suitable, Relatively suitable, Moderately suitable, Slightly suitable, and Unsuitable (as shown in Fig 6). The specific conditions of each category are detailed in Table 4.

In terms of latitude, the suitability is higher in the eastern part of the urban area and lower in the central and western regions. Longitudinally, the central urban area shows lower suitability, while the northern and southern regions exhibit higher suitability. Overall, the unsuitable areas are mostly in regions with high human activity, low vegetation cover, and dense urban road networks. Therefore, noise control measures in Nanchang should focus on these areas. A coordinated effort by the government and the public, involving source management, improved urban planning and design, enhanced environmental monitoring and assessment, and comprehensive legislation, is essential to effectively address urban noise pollution.

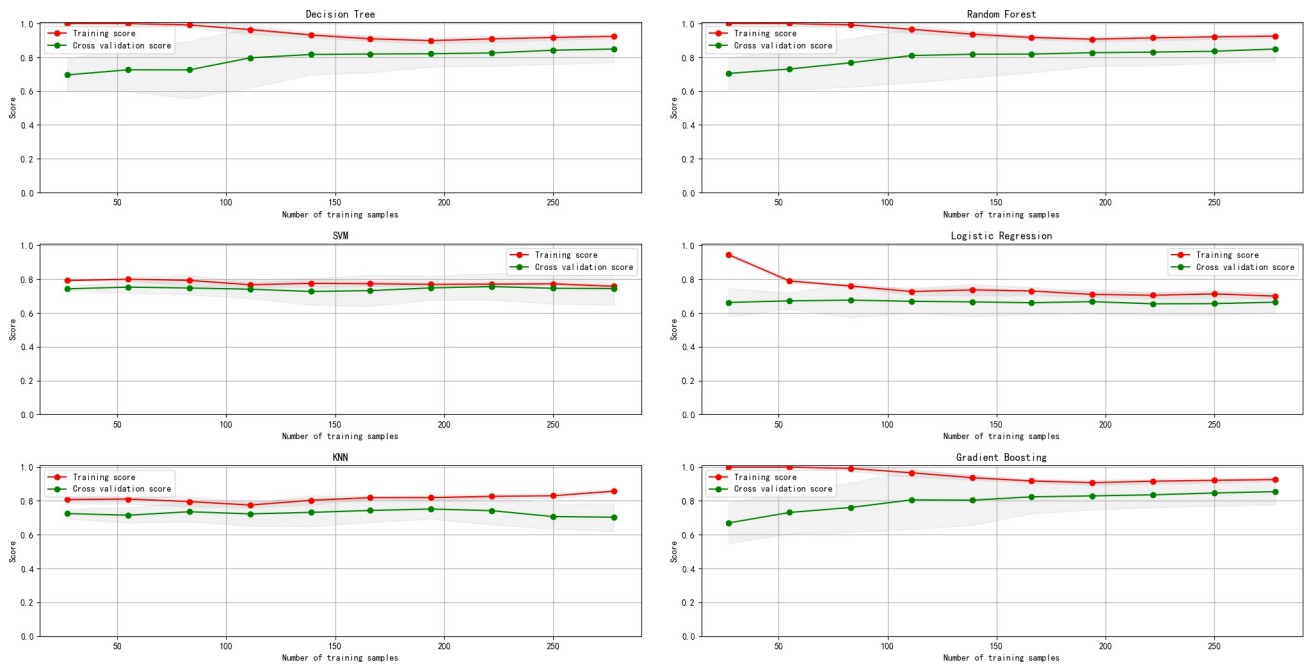

**Fig 5. Learning curves under various machine algorithms.**

### 3.3 Analysis of driving factors

Analyzing the feature importance results returned by the model (Fig 7) reveals that both urban development conditions and natural environmental conditions significantly impact the noise suitability model. Among these, population spatial distribution has the greatest influence, indicating it as a primary source of urban noise. Therefore, adjusting the population distribution can mitigate the negative effects of urban noise and improve the quality of the urban living environment.

The second most influential factors are the distance to water bodies and road network density. This is closely related to Nanchang's "one river, ten lakes" urban water network layout, where areas near water often have high population density and frequent human activities. Large open water bodies also facilitate sound propagation and scattering. Road network density reflects the richness and complexity of an area's transportation infrastructure, often indicating a higher level of urbanization. Consequently, both factors significantly contribute to urban noise levels.

Moderately influential features include vegetation cover, which reflects the distribution of vegetation in the city and can effectively reduce noise pollution, improving the overall acoustic environment. Environmental purification value indicates the city's natural or artificial noise reduction capacity and has a moderate impact on urban noise. Large vehicle traffic volume refers to the hourly flow of large vehicles, which typically generate more noise than smaller vehicles. However, city management measures for large vehicles can reduce the expected impact of this feature. Biodiversity value reflects species diversity within the urban ecosystem, with higher biodiversity often indicating favorable natural conditions that can indirectly reduce noise.

Annual average wind speed affects noise propagation and can increase background noise levels due to terrain and building layout. Elevation reflects the topography, with different elevation areas exhibiting varying sound propagation characteristics. Distance to residential areas

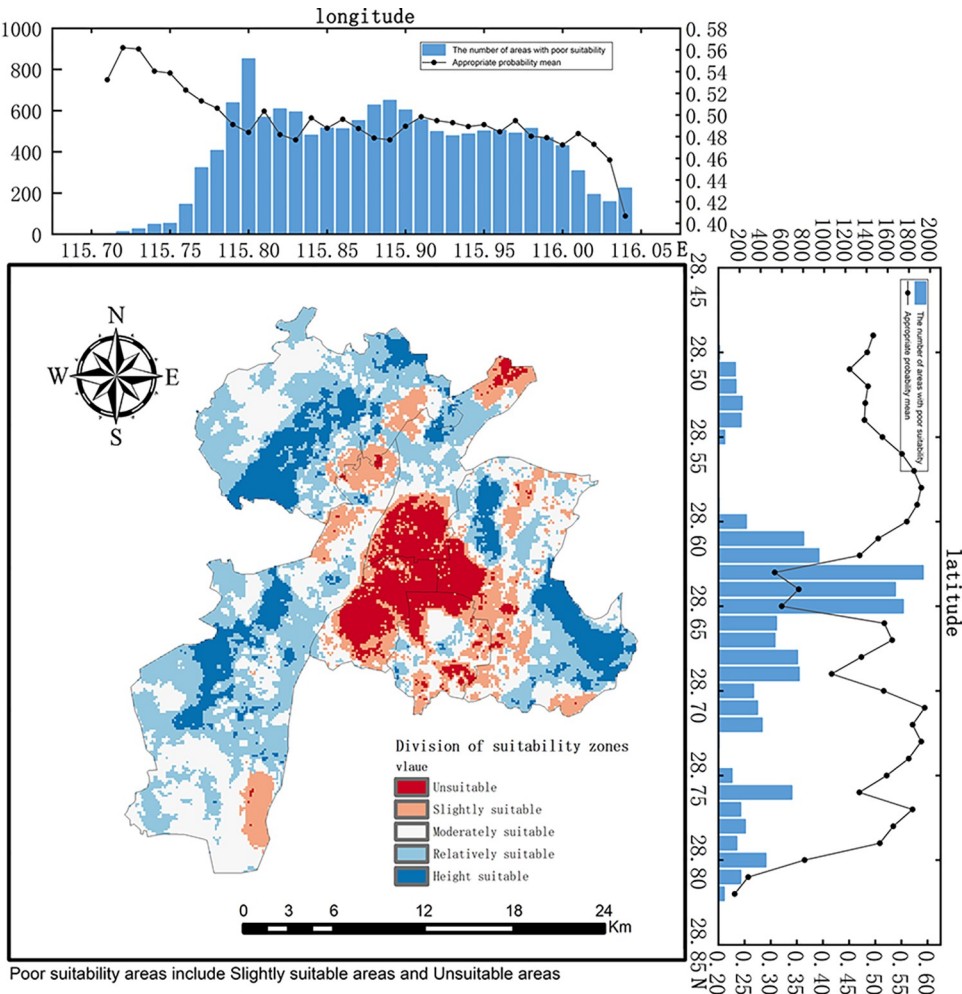

**Fig 6. Classification of noise suitability areas in the central urban area.**

indicates the proximity to residential points, reflecting the frequency of human activities, and thus impacts urban noise to some extent. Features with lower contributions, such as land use type and building height, provide limited heterogeneity for the model to learn from, resulting in lower contribution rates to the model.

**Table 4. Classification of noise suitability areas in the central urban area.**

| | Appropriate probability | Area km² | Proportion % | Distribution |
|---|---|---|---|---|
| Unsuitable areas | 0.01–0.2168 | 54.9 | 9.38 | Mainly distributed in the central and eastern regions of urban areas |
| Slightly suitable areas | 0.2168–0.3910 | 93.79 | 16.03 | Distributed in the eastern part of the urban area, along both sides of the Gan River |
| Moderately suitable areas | 0.3910–0.5653 | 163.96 | 28.02 | Widely distributed in the northwest and southwest of the urban area |
| Relatively suitable areas | 0.5653–0.7396 | 194.88 | 33.31 | Mainly distributed in the western part of the urban area, with a few areas in the eastern part |
| Height suitable areas | 0.7396–0.96 | 77.55 | 13.25 | Concentrated in the eastern, western, and northwestern regions |

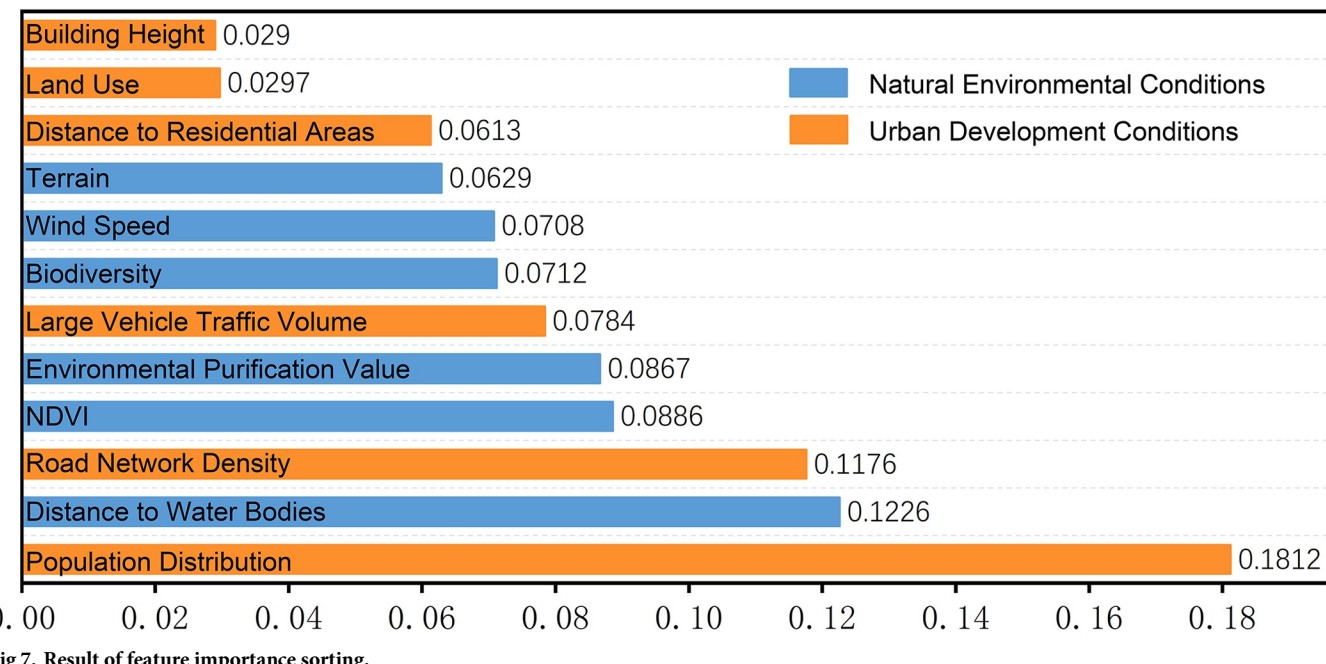

**Fig 7. Result of feature importance sorting.**

## 3.4 SHAP model interpretation

In machine learning and deep learning, model interpretability is a crucial topic. Despite the excellent predictive performance of complex models such as deep neural networks and ensemble models, they are often considered "black boxes" due to the difficulty in explaining their internal decision-making processes. SHAP (SHapley Additive exPlanations) is a tool designed to address this issue by assigning importance values to features to explain the model's output [37]. The core idea of SHAP is derived from the Shapley value in cooperative game theory, which is used to fairly distribute the gains among participants in a coalition [39]. SHAP applies this concept to machine learning to calculate each feature's contribution to the model's prediction.

Using SHAP dependence plots, SHAP can also reveal the interactions between individual features. Fig 8 illustrates the dependence plots for the distance to water bodies, population spatial distribution, and road density. These three features demonstrate higher importance compared to others. The dependence plots highlight the contribution of these features to the

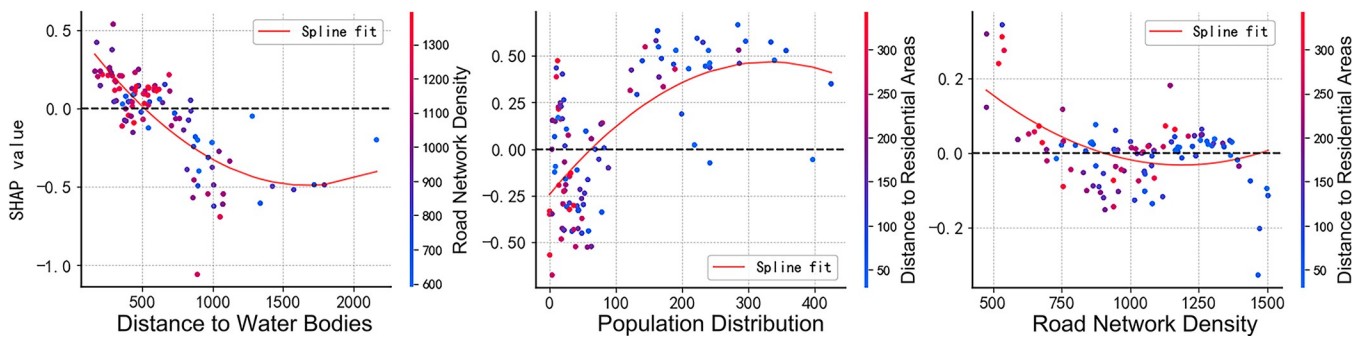

**Fig 8. SHAP dependency graph.**

output and identify the features with which they primarily interact. The SHAP value for noise is highest when the distance to water bodies is within 500 meters and reaches its lowest point at approximately 800 meters. The color scale on the right indicates the variable with which the distance to water bodies mainly interacts. As shown in the figure, the random forest model suggests that the distance to water bodies is primarily associated with road network density, and the SHAP value increases significantly when the distance interacts with higher road network density (represented by red dots).

Regarding population spatial distribution, the SHAP value for noise reaches its maximum when the population spatial distribution exceeds 150, while lower SHAP values are observed within the range of 100. The model indicates a strong correlation between population spatial distribution and distance from residential areas. Most points with a greater distance from residential areas (red dots) are concentrated within the population spatial distribution range of 100. A greater distance from residential areas combined with a smaller population distribution typically results in a lower urban noise environment.

For road network density, the SHAP value for noise peaks around a density of 500 and reaches its lowest values between 750 and 1125. Similar to population spatial distribution, the model suggests a strong correlation between road network density and distance from residential areas.

## 4 Discussion

The potential benefits of machine learning are increasingly recognized by the public. Compared to traditional methods, machine learning offers faster results [40], which is particularly important for the rapidly changing urban noise environment. In this study, we evaluated urban noise using the Random Forest algorithm, which was selected after comparing various algorithms based on relevant parameters. This approach eliminates the need for costly on-site measurements. The accuracy of the model depends not only on the choice of the correct algorithm but also on the availability of extensive data [41]. By aggregating multi-source geospatial data, we fully utilized existing resources, thereby overcoming cost and time constraints. The final results demonstrate high model accuracy and identify key features influencing noise distribution, particularly population spatial distribution, distance to water bodies, and road network density. This confirms the robust capability of machine learning models in handling complex and nonlinear data, especially in environmental science involving geospatial data.

As a low-cost, efficient, and accurate approach for urban noise suitability evaluation, this study not only innovates noise assessment methods but also provides scientific decision-making support for urban planning and environmental management [42]. Effective noise control strategies are crucial for improving residents' quality of life and achieving sustainable urban development, especially in the context of rapid urbanization [43]. By identifying and prioritizing key geographical factors influencing noise distribution, policymakers and planners can implement targeted noise reduction measures. As the Nanchang municipal government continues to deploy automatic noise monitoring stations, the study's database can be expanded in the future to include 24-hour monitoring data from these fixed points. This will allow for more detailed and in-depth research by incorporating the temporal dimension of noise variations.

## 5 Conclusion

This study uses machine learning to evaluate urban noise suitability by assessing the accuracy of six machine learning algorithms. The Random Forest algorithm was selected for empirical research on urban noise suitability, yielding the following conclusions:

The machine learning-based urban noise suitability evaluation model, leveraging data mining and machine learning technology, identifies potential patterns and trends from existing data samples. It provides a more economical, efficient, and reliable assessment of urban noise suitability and serves as a supplementary approach for noise control by urban managers. Among the six machine learning models, the Random Forest algorithm exhibited the best performance with a Test F1 Score of 0.8636, a Training F1 Score of 0.9218, and a high AUC of 0.90. The model achieved a good balance between overall classification accuracy and recall, demonstrating excellent performance.

According to the model's predictions, areas with low noise suitability are concentrated in the city center, while the surrounding areas show higher noise suitability. The areas classified as Unsuitable, Slightly suitable, Moderately suitable, Relatively suitable, and Height suitable account for 9.38%, 16.03%, 28.02%, 33.31%, and 13.25% of the central urban area, respectively.

The feature importance ranking indicates that population spatial distribution has the highest contribution, making it the primary factor influencing urban noise. Distance to water bodies and road network density are also significant factors. According to the SHAP dependence plots, when the population spatial distribution exceeds 150, the SHAP value for noise reaches its maximum, while lower SHAP values are observed within the range of 100. Similarly, the SHAP value for noise is highest when the distance to water bodies is within 500 meters and lowest around 800 meters. Additionally, when distance to water bodies interacts with higher road network density, the SHAP value is elevated. For road network density, the SHAP value for noise reaches its maximum around a density of 500 units and its minimum between 750 and 1125 units.

## Supporting information

**S1 Data.**
(ZIP)

## Author Contributions

**Conceptualization:** Jinlin Teng, Chunqing Liu.

**Data curation:** Cheng Zhang.

**Investigation:** Cheng Zhang.

**Methodology:** Huimin Gong.

**Project administration:** Huimin Gong.

**Resources:** Huimin Gong.

**Software:** Jinlin Teng.

**Supervision:** Chunqing Liu.

**Validation:** Cheng Zhang.

**Visualization:** Jinlin Teng.

**Writing – original draft:** Jinlin Teng.

**Writing – review & editing:** Huimin Gong.

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
