## [Decision Letter · Decision Letter 0]

15 Aug 2024

PONE-D-24-29106

Machine Learning-based Urban Noise Appropriateness Evaluation Method and Driving Factor Analysis

PLOS ONE

Dear Dr. Teng,

Thank you for submitting your manuscript to PLOS ONE. After careful consideration, we feel that it has merit but does not fully meet PLOS ONE’s publication criteria as it currently stands. Therefore, we invite you to submit a revised version of the manuscript that addresses the points raised during the review process.

We look forward to receiving your revised manuscript.

Kind regards,

Upaka Rathnayake, PhD

Academic Editor

PLOS ONE

Journal Requirements:

3. We note that your Data Availability Statement is currently as follows: All relevant data are within the manuscript and its Supporting Information files

4. We note that Figures 1,2 and 6 in your submission contain map/satellite images which may be copyrighted. All PLOS content is published under the Creative Commons Attribution License (CC BY 4.0), which means that the manuscript, images, and Supporting Information files will be freely available online, and any third party is permitted to access, download, copy, distribute, and use these materials in any way, even commercially, with proper attribution. For these reasons, we cannot publish previously copyrighted maps or satellite images created using proprietary data, such as Google software (Google Maps, Street View, and Earth). For more information, see our copyright guidelines: http://journals.plos.org/plosone/s/licenses-and-copyright.

a. You may seek permission from the original copyright holder of Figures 1,2 and 6 to publish the content specifically under the CC BY 4.0 license.  

Reviewers' comments:

Reviewer's Responses to Questions

**Comments to the Author**

1. Is the manuscript technically sound, and do the data support the conclusions?

Reviewer #1: Yes

Reviewer #2: Partly

2. Has the statistical analysis been performed appropriately and rigorously? 

Reviewer #1: Yes

Reviewer #2: Yes

3. Have the authors made all data underlying the findings in their manuscript fully available?

Reviewer #1: Yes

Reviewer #2: Yes

4. Is the manuscript presented in an intelligible fashion and written in standard English?

Reviewer #1: Yes

Reviewer #2: Yes

5. Review Comments to the Author

Reviewer #1: Good study and i really appreciate the presentation of results in an interesting way.

FOllowing points can be revised in the manuscript

1. "However, interpolated data sometimes fail to accurately reflect actual noise levels in certain local areas.

This paper proposes a machine learning-based solution for urban " Here the flow is not there. Need to flow your ideas logically, why you need machine learning.

2. Any reason for the particular selection of the study area ?

3. If possible improve the clarity of Figure 1

4. Why did you choose these algorithms ? Please justify. FOr your reference, these classical machine learning methods have been widely used in engineering applications

https://www.sciencedirect.com/science/article/pii/S2590123023005157

https://www.nature.com/articles/s41598-023-40513-x

5. "models such as deep neural networks and ensemble

models, they are often considered "black boxes" due to the difficulty in explaining their internal

decision-making processes" For your reference : https://www.sciencedirect.com/science/article/pii/S2590123024007588

6. You might need to revisit the dependency plots description. The right side y axis is showing only the next best variable which is mostly associated with the feature in x axis, It is an optional plot.. Main dependency plot is the x axis and y (left) axis. Colour bar comes with that second feature (mostly associated). The present version is bit confusing

Reviewer #2: Though work have some novelty but many limitations which made me some reservations.

Before conclusions authors have mentioned line 303

However, this study has several limitations and areas for improvement , how this justify novelty and readers may have to understand this.

Authors should refer recent past related papers to improve the discussions

Analysis of Data Splitting on Streamflow Prediction using Random Forest (aimspress.com)

AIMS Environmental Science

2024, Volume 11, Issue 4: 593-609. doi: 10.3934/environsci.2024029

Previous ArticleNext Article

Kumar, V.; Azamathulla, H.M.; Sharma, K.V.; Mehta, D.J.; Maharaj, K.T. The State of the Art in Deep Learning Applications, Challenges, and Future Prospects: A Comprehensive Review of Flood Forecasting and Management. Sustainability 2023, 15, 10543. doi: 10.3390/su151310543

Mahtabi, G.; ; Pal, M. Classification of Hydraulic Jump in Rough Beds. Water 2020, 12, 2249. doi: 10.3390/w12082249

6. PLOS authors have the option to publish the peer review history of their article (what does this mean?). If published, this will include your full peer review and any attached files.

Reviewer #1: No

Reviewer #2: **Yes: **Mohammad Azamathulla Hazi

---

## [Author Response · Author response to Decision Letter 0]

15 Sep 2024

Dear Dr. Upaka Rathnayake and Dr. Paula Katrina A. Maderazo,

I hope this email finds you well. We sincerely appreciate your comprehensive review of our manuscript. Based on your valuable feedback, particularly your request to provide a new link to the Chinese Standard Map Service System, we have made the following explanations and modifications:

1.Source of Maps: Since the Standard Map Service of China is the official website of the Ministry of Natural Resources of China, it may be subject to geographical restrictions and network policies. As a result, accessing it from abroad could lead to issues such as slow loading times, failure to load, or complete blockage. We have therefore revisited the images in the manuscript that may involve copyrighted material (Figures 1, 2, and 6). We have removed any satellite or map images that could raise copyright concerns and replaced them with open-access maps from sources compliant with the Plos One copyright guidelines. The details can be found in Table 2 of the "Revised Manuscript with Track Changes." A brief explanation is as follows:

①Figure 1 does not involve satellite maps. The vector map of roads, water bodies, green spaces, and other elements is sourced from the public dataset available at OpenStreetMap.（www.openstreetmap.org）

② Figure 2 is based on various data sources. We carefully reviewed the origin of each subfigure, which includes: Land Use Data, Population Data, Remote Sensing Data, Ecosystem Service Value Data, Meteorological Data, Basic Vector Data, Building Height Data, Residential POI Data, and Noise Monitoring Data. Accordingly, we have updated Table 2 to reflect these sources. Notably:

"Remote Sensing Data" has been changed from Geospatial Data Cloud to "USGS Remote Sensing Data". Using Landsat 8 OLI-TIRS data for calculating Normalized Difference Vegetation Index (NDVI)（https://earthexplorer.usgs.gov/）

"Basic Vector Data" is now sourced from the public dataset at OpenStreetMap.（www.openstreetmap.org）

"Building Height Data" is sourced from the public dataset available at the Zenodo Platform Database, accessible via this link：https://zenodo.org/records/7827315

"Population Data" is sourced from the public WORLDPOP Dataset, available at www.worldpop.org.

"Residential POI Data" is obtained from the Baidu Map API Open Interface, available at this link：https://lbsyun.baidu.com/products/advantage?active=service.

Additionally, "Ecosystem Service Value Data" and "Meteorological Data" are sourced from the public Resource and Environmental Science Data Platform, accessible at http://www.resdc.cn/DOI.

"Noise Monitoring Data" is sourced from the Nanchang Environmental Protection Bureau Environmental Quality Monitoring Database, available at http://sthjj.nc.gov.cn/ncgbj/zsjcbg/nav_list.shtml.

③ Figure 6 is based on the data and content described earlier in the manuscript. The final prediction results were obtained using machine learning techniques and do not involve any third-party copyright concerns. This figure represents one of the key findings of our study.

2．Manuscript Formatting: I have carefully reviewed and ensured that the manuscript now fully complies with the formatting guidelines of PLOS ONE.

3．Data and Code Availability: We have uploaded the code and urban noise monitoring data used in the study to GitHub. This repository also includes the final prediction data, supporting the principles of reproducibility and open data. You can access the repository at the following link: https://github.com/tengjinlin/noise-prediction.git. 

4．Image Conversion: Lastly, all images in the manuscript have been converted using the PACE conversion engine's digital diagnostic tool (available at https://pacev2.apexcovantage.com/) to ensure they meet the image quality requirements of PLOS.

Thank you again for your time and effort in reviewing our work. We hope these revisions address your concerns, and we look forward to your feedback.

Best regards,

TENG Jinlin

College of Forestry, Jiangxi Agricultural University, Nanchang 330045, Jiangxi Province，China

E-mail:tengjinlin@stu.jxau.edu.cn

Dear Reviewer 1,

Thank you for your valuable feedback. We have carefully considered your comments and have made the following revisions to the manuscript:

1．Introduction logic of research methods: We have revised the manuscript to clarify that, although the Kriging interpolation method accounts for the spatial correlation of spatial data and can sometimes produce accurate predictions, it may be significantly impacted by uneven sample point distribution or outliers in certain areas. This study, therefore, proposes a machine learning approach, which relies on large datasets for training and learning, enabling it to automatically extract features and patterns from the data. This means that even if the interpolation method fails in some localized areas, the machine learning model can still predict the actual levels in those areas by learning from data in other regions. The revisions can be found in the “Revised Manuscript with Track Changes.”

2．Rationale for Selecting Nanchang City: We chose Nanchang City as our study area because it is the capital of Jiangxi Province and represents typical noise environments found in Chinese provincial capital cities. Like other large and medium-sized cities, Nanchang faces significant population pressures, leading to common issues such as traffic noise, industrial noise, construction noise, and noise from social activities. Therefore, Nanchang serves as a representative case for studying noise environments in similar urban settings.

3．Image Quality: We have improved the clarity of the figures in the manuscript as per your suggestion. The updated figures are attached separately in this email.

4．Explanation for Choosing the Six Algorithms: Based on the references you provided and additional literature retrieved from databases, we have elaborated on the details of the six algorithms selected for the study and explained the rationale behind their selection.

Decision Trees (DT) are tree-based algorithms used for classification and regression tasks. They construct a tree structure by dividing the dataset into subsets, with each node representing a feature, each branch representing a feature value, and the final leaf nodes representing a class (or regression value). Decision trees are easy to understand and interpret, and they can handle both numerical and categorical data, but they are prone to overfitting.

Random Forest (RF) is an ensemble learning algorithm that improves the accuracy of classification and regression tasks by training multiple decision trees. It enhances model performance by averaging or voting on the predictions of numerous decision trees, reducing overfitting. RF is suitable for large datasets, robust to outliers and missing data, and can estimate feature importance.

Support Vector Machine (SVM) is a supervised learning algorithm used for binary classification and regression tasks. It finds the optimal hyperplane in the feature space to separate data points from different classes, maximizing the margin between them. SVM has highly flexible kernel functions, can handle high-dimensional datasets, and is robust to outliers, but it requires longer training times for large datasets.

Logistic Regression is a linear model used for binary classification problems. It classifies based on a linear combination of features, using the logistic function to convert the linear output into probabilities, which are then used for classification based on a threshold. Logistic regression is simple and easy to implement, can output class probabilities, but has limited ability to model non-linear problems.

K-Nearest Neighbors (KNN) is an instance-based learning algorithm used for classification and regression tasks. It classifies by considering the class of the nearest neighbors to the input instance, using voting to determine the class of a sample. KNN requires no model training, is easy to understand and implement, but has slower prediction speed for large datasets and is less suitable for high-dimensional data.

Gradient Boosting is an ensemble learning algorithm that improves model performance by sequentially training multiple weak learners. It iteratively trains new models to correct the errors of the previous model, gradually enhancing model performance. Gradient boosting generally performs well in prediction tasks, but it is sensitive to hyperparameter selection and tuning, requiring some experience in hyperparameter optimization.

5．Model Interpretability: We have incorporated the Shapley Additive Explanations (SHAP) method to enhance the interpretability of the model, as per your suggestion. We specifically referenced the following paper:

Kulasooriya WKVJ, Ranasinghe RSS, Perera US, Thisovithan P, Ekanayake IU, Meddage DPP. "Modeling strength characteristics of basalt fiber reinforced concrete using multiple explainable machine learning with a graphical user interface." Scientific Reports. 2023; 13(1):13138. Available from: https://go.exlibris.link/ckWy5VZt doi: 10.1038/s41598-023-40513-x.

6．Revised SHAP Section: I have restructured the section on SHAP explanations to improve its clarity and readability. Below is the revised text:

Using SHAP dependence plots, SHAP can also reveal the interactions between individual features. Figure 8 illustrates the dependence plots for the distance to water bodies, population spatial distribution, and road density. These three features demonstrate higher importance compared to others. The dependence plots highlight the contribution of these features to the output and identify the features with which they primarily interact. The SHAP value for noise is highest when the distance to water bodies is within 500 meters and reaches its lowest point at approximately 800 meters. The color scale on the right indicates the variable with which the distance to water bodies mainly interacts. As shown in the figure, the random forest model suggests that the distance to water bodies is primarily associated with road network density, and the SHAP value increases significantly when the distance interacts with higher road network density (represented by red dots).

Regarding population spatial distribution, the SHAP value for noise reaches its maximum when the population spatial distribution exceeds 150, while lower SHAP values are observed within the range of 100. The model indicates a strong correlation between population spatial distribution and distance from residential areas. Most points with a greater distance from residential areas (red dots) are concentrated within the population spatial distribution range of 100. A greater distance from residential areas combined with a smaller population distribution typically results in a lower urban noise environment.

For road network density, the SHAP value for noise peaks around a density of 500 and reaches its lowest values between 750 and 1125. Similar to population spatial distribution, the model suggests a strong correlation between road network density and distance from residential areas.

Thank you once again for your valuable feedback. We hope these revisions meet your expectations, and we look forward to any further suggestions you may have.

Best regards,

TENG Jinlin

College of Forestry, Jiangxi Agricultural University, Nanchang 330045, Jiangxi Province，China

E-mail:tengjinlin@stu.jxau.edu.cn

Dear Reviewer 2,

Dear Dr. Mohammad Azamathulla Hazi,

Thank you very much for your valuable feedback. Your suggestions, particularly the three reference papers you provided, have significantly enhanced the originality of the manuscript and the writing of the discussion section. Below is a summary of our responses and the corresponding revisions made to the manuscript:

The potential benefits of machine learning are increasingly recognized by the public. Compared to traditional methods, machine learning offers faster results[40], which is particularly important for the rapidly changing urban noise environment. In this study, we evaluated urban noise using the Random Forest algorithm, which was selected after comparing various algorithms based on relevant parameters. This approach eliminates the need for costly on-site measurements. The accuracy of the model depends not only on the choice of the correct algorithm but also on the availability of extensive data[41]. By aggregating multi-source geospatial data, we fully utilized existing resources, thereby overcoming cost and time constraints. The final results demonstrate high model accuracy and identify key features influencing noise distribution, particularly population spatial distribution, distance to water bodies, and road network density. This confirms the robust capability of machine learning models in handling complex and nonlinear data, especially in environmental science involving geospatial data.

As a low-cost, efficient, and accurate approach for urban noise suitability evaluation, this study not only innovates noise assessment methods but also provides scientific decision-making support for urban planning and environmental management[42]. Effective noise control strategies are crucial for improving residents' quality of life and achieving sustainable urban development, especially in the context of rapid urbanization[43]. By identifying and prioritizing key geographical factors influencing noise distribution, policymakers and planners can implement targeted noise reduction measures. As the Nanchang municipal government continues to deploy automatic noise monitoring stations, the study's database can be expanded in the future to include 24-hour monitoring data from these fixed points. This will allow for more detailed and in-depth research by incorporating the temporal dimension of noise variations.

Thank you once again for your valuable feedback. We hope these revisions meet your expectations, and we look forward to any further suggestions you may have.

Best regards,

TENG Jinlin

College of Forestry, Jiangxi Agricultural University, Nanchang 330045, Jiangxi Province，China

E-mail:tengjinlin@stu.jxau.edu.cn

---

## [Decision Letter · Decision Letter 1]

23 Sep 2024

Machine Learning-based Urban Noise Appropriateness Evaluation Method and Driving Factor Analysis

PONE-D-24-29106R1

Dear Dr. Chunqing,

We’re pleased to inform you that your manuscript has been judged scientifically suitable for publication and will be formally accepted for publication once it meets all outstanding technical requirements.

Kind regards,

Upaka Rathnayake, PhD

Academic Editor

PLOS ONE

Additional Editor Comments (optional):

Reviewers' comments:

Reviewer's Responses to Questions

**Comments to the Author**

1. If the authors have adequately addressed your comments raised in a previous round of review and you feel that this manuscript is now acceptable for publication, you may indicate that here to bypass the “Comments to the Author” section, enter your conflict of interest statement in the “Confidential to Editor” section, and submit your "Accept" recommendation.

Reviewer #1: All comments have been addressed

Reviewer #2: All comments have been addressed

2. Is the manuscript technically sound, and do the data support the conclusions?

Reviewer #1: Yes

Reviewer #2: Yes

3. Has the statistical analysis been performed appropriately and rigorously? 

Reviewer #1: Yes

Reviewer #2: Yes

4. Have the authors made all data underlying the findings in their manuscript fully available?

Reviewer #1: No

Reviewer #2: Yes

5. Is the manuscript presented in an intelligible fashion and written in standard English?

Reviewer #1: Yes

Reviewer #2: Yes

6. Review Comments to the Author

Reviewer #1: THe authors have successfully addressed the comments. The manuscript is acceptable.

Reviewer #2: Manuscript Number PONE-D-24-29106R1

Machine Learning-based Urban Noise Appropriateness Evaluation Method and Driving Factor Analysis

Manuscript Number PONE-D-24-29106R1

Machine Learning-based Urban Noise Appropriateness Evaluation Method and Driving Factor Analysis

Dear authors

thanks for applying all corrections

7. PLOS authors have the option to publish the peer review history of their article (what does this mean?). If published, this will include your full peer review and any attached files.

Reviewer #1: No

Reviewer #2: **Yes: **Mohammad Azamathulla Hazi

---

## [Editor Report · Acceptance letter]

11 Dec 2024

PONE-D-24-29106R1 

PLOS ONE

Dear Dr. Liu, 

I'm pleased to inform you that your manuscript has been deemed suitable for publication in PLOS ONE. Congratulations! Your manuscript is now being handed over to our production team.

Kind regards, 

on behalf of

Professor Upaka Rathnayake 

Academic Editor

PLOS ONE